

# Nanopublication beyond the sciences: the PeriodO period gazetteer

Patrick Golden and Ryan Shaw

School of Information and Library Science, University of North Carolina at Chapel Hill, Chapel Hill, NC, United States

## ABSTRACT

The information expressed in humanities datasets is inextricably tied to a wider discursive environment that is irreducible to complete formal representation. Humanities scholars must wrestle with this fact when they attempt to publish or consume structured data. The practice of "nanopublication," which originated in the e-science domain, offers a way to maintain the connection between formal representations of humanities data and its discursive basis. In this paper we describe nanopublication, its potential applicability to the humanities, and our experience curating humanities nanopublications in the PeriodO period gazetteer.

## INTRODUCTION

Humanities scholars who wish to make their research materials usable with networked digital tools face a common dilemma: How can one publish research materials as "data" without severing them from the ideas and texts that originally gave them meaning? The kinds of information produced in the humanities—biographical details, political and temporal boundaries, and relationships between people, places, and events—are inextricably tied to arguments made by humanities scholars. Converting all, or even much, of the information expressed in scholarly discourse into algorithmically processable chunks of formal, structured data has so far proven to be extraordinarily difficult.

But rather than attempt to exhaustively represent her research, a scholar can promote small pieces of information within her work using the practice of *nanopublication* (*Mons & Velterop, 2009*). Nanopublications include useful and usable representations of the provenance of structured assertions. These representations of provenance are useful because they allow consumers of the published data to make connections to other sources of information about the context of the production of that data. In this way, they strike a balance between the needs of computers for uniformity in data modeling with the needs of humans to judge information based on the wider context of its production. An emphasis on connecting assertions with their authors is particularly well-suited for the needs of humanities scholars. By adopting nanopublication, creators of datasets in the humanities can focus on publishing small units of practically useful, curated assertions while keeping a persistent pointer to the basis of those claims—the discourse of scholarly publishing itself—rather than its isolated representation in formal logic.

Corresponding author
Patrick Golden,
ptgolden@email.unc.edu

We offer as an example of this approach the PeriodO period gazetteer, which collects definitions of time periods made by archaeologists and other historical scholars (http://perio.do). A major goal of the gazetteer was to make period definitions parsable and comparable by computers, while also retaining links to the broader scholarly context in which they were conceived. We found that a nanopublication-centric approach allowed us to achieve this goal. In this paper, we describe the concept of nanopublication, its origin in the hard sciences, and its applicability to the humanities. We then describe the PeriodO period gazetteer in detail, discuss our experience mapping nonscientific data into nanopublications, and offer advice to other humanities-oriented projects attempting to do the same.

## NANOPUBLICATIONS

Nanopublication is an approach to publishing research in which individual research findings are modeled as structured data in such a way that they retain information about their provenance. This is in contrast to both traditional narrative publishing, where research findings are not typically published in a structured, computer readable format, and "data dumps" of research findings which are typically published without any embedded information about their origin or production. The nanopublication approach is motivated by a desire to publish structured data without losing the wider research context and the benefits of traditional scholarly communication (*Groth, Gibson & Velterop, 2010*).

Nanopublication emerged from work in data-intensive sciences like genomics and bioinformatics, where recent advances in computational measurement techniques have vastly lowered the barrier to collecting genetic sequencing data. As a result, millions of papers have been published with findings based on these new methods. However, the reported results are almost always published in the form of traditional narrative scholarly publications (*Mons et al., 2011*). While narrative results can be read and understood by humans, they are not so easily digested by computers. In fields where computation has been the key to the ability to ask new and broader questions, it should surely be the case that research results are published in such a way that they are able to be easily parsed, collected, and compared by computer programs and the researchers who use them.

On the occasions when research data are released and shared, they are often distributed on their own, stripped of the context necessary to locate them within a broad research environment (the identity of the researchers, where and how this research was conducted, etc.). In this case, publishing practice has swung too far to the opposite extreme. In the service of creating and sharing discrete datasets, the published results have been stripped of their provenance and their position within the wider scholarly endeavor that culminated in their publication. This contextual information is crucial for researchers to determine the trustworthiness of the dataset and learn about the broader project of research from which they resulted.

Nanopublication offers a supplementary form of publishing alongside traditional narrative publications. A nanopublication consists of three parts, all representable by RDF graphs:

1. An assertion (a small, unambiguous unit of information)
2. The provenance of that assertion (who made that assertion, where, when, etc.)
3. The provenance of the nanopublication itself (who formed or extracted the assertion, when, and by what method)

The formal definitions of these parts are specified by an OWL ontology (*Groth et al., 2013*). By representing their research in nanopublications alongside their narrative reports, researchers can publish their data in such a way that the data remain within their human context while also being easily digested by computer programs.

Authors are encouraged to include the smallest possible unambiguous pieces of information as the assertions at the center of a nanopublication. In the bioscience context, these assertions could range from statements of causality, to measurements of gene expressions or gene-disease associations, to statistics about drug interactions. The scope and nature of appropriate units of nanopublication inevitably vary by discipline. Multiple statements of identical or closely related facts can be connected with different sources of provenance, thereby potentially augmenting the ability of consumers to judge the quality of assertions. *Groth, Gibson & Velterop (2010)* call the collection of nanopublications all referring to the same assertion "S-evidence," and cite the potential benefits of the ability to automatically connect findings across research publications.

Several European repositories of bioinformatic data have begun to publish their contents as nanopublications, including the Biosemantics Group (http://www.biosemantics.org), neXtProt (http://nextprot.org/), and DisGeNET (http://www.disgenet.org/web/DisGeNET/v2.1). These publications can be aggregated and connected in larger systems, such as the decentralized reputation system described by *Kuhn (2015)*.

## NANOPUBLICATION IN THE HUMANITIES

While the bioinformatics research community has enthusiastically adopted nanopublication, other disciplines have been slow to follow. *Gradmann (2014)* suggested that specialized and stable terminologies, as well as sufficient funding to organize these terminologies in formal ontologies, may be prerequisites for the successful deployment of nanopublication. Thus while he expects other scientific, technical, and medical disciplines to eventually embrace nanopublication, he is less sure that nanopublication will work for the humanities. Historians, for example, use relatively little specialized terminology and pride themselves on their ability to use "ordinary language" to represent the past. Even when humanities scholars use specialized theoretical language, their use of this language is often unstable, ambiguous, and highly contested. Perhaps, then, a publishing technique that seeks to eliminate such ambiguity is ill-suited for these fields.

A related obstacle to the adoption of nanopublication beyond the hard sciences has to do with differences in the role played by "facts." Researchers trained in the hard sciences understand their work to be cumulative: scientists "stand on the shoulders of giants" and build upon the work of earlier researchers. While scientists can in principle go back and recreate the experiments of their predecessors, in practice they do this only when the results

 

of those experiments have not been sufficiently established as facts. Efficient cumulative research requires that, most of the time, they simply trust that the facts they inherit work as advertised. Something like this process seems to be assumed by many proponents of nanopublications. For example, *Mons & Velterop (2009)* claim that a major goal of nanopublication is to "elevate" factual observations made by scientists into standardized packages that can be accumulated in databases, at least until they are proved wrong. These standardized packages can then be automatically or semi-automatically analyzed to produce new factual observations (or hypotheses about potential observations), and the cycle continues.

Yet as *Mink (1966)* observed, not all forms of research and scholarship are aimed at producing "detachable conclusions" that can serve as the basis for a cumulative process of knowledge production. Anticipating Gradmann, Mink argued that

> Detachable conclusions are possible in science because—and only because—of its theoretical structure. The division of labor in research requires that concepts have a uniformity of meaning, and the methodological problem of definition therefore becomes central (*Mink, 1966*, 39).

Mink contrasted science to the study of history, which, lacking both explicit methodology and uniform consensus on the meanings of its concepts, does not produce "detachable conclusions." But this does not mean that historical scholarship fails to produce knowledge, only that it is a separate and autonomous mode of understanding. The goal of most historical scholarship is not to establish conclusions by constructing an explanatory chain of inferences from evidence. Rather the goal is to render what Mink called a "synoptic judgment," an interpretive act in which the scholar comes to "see together" the disparate observable elements of some phenomena as a synthetic whole. The historian who judges the advent of printing to have constituted a "communications revolution" (*Eisenstein, 1979*) has not made an inference from the available evidence but has constructed a particular interpretation of that evidence. To communicate her synoptic judgment to others, she cannot simply state her conclusions unambiguously and rely on her audience's theoretical understanding to make them meaningful; instead she must arrange and exhibit the evidence to help them "see together" what she saw.

So is nanopublication a poor fit for fields of knowledge production that do not follow the model of cumulative science? We believe the answer is no. First of all, even Mink did not argue that there were no facts in history, only that the significant conclusions drawn by historians do not typically take the form of factual statements. There are plenty of equivalents in history and the humanities to the databases of curated factual statements that exists in the sciences: prosopographical databases (*Bradley & Short, 2005*), digital historical gazetteers (*Elliott & Gillies, 2011*), not to mention the catalogs and indexes of bibliographical data that make humanities scholarship possible (*Buckland, 2006*). Some of these facts may be vague or uncertain, but as *Kuhn et al. (2013)* observe, even knowledge that cannot be completely formally represented, including vague or uncertain scientific findings, can benefit from the nanopublication approach. We agree but would go further

to say that nanopublication is useful even for information that is neither testable nor falsifiable, exemplified by Mink's synoptic judgments. We have demonstrated the utility of nanopublications for describing synoptic judgments of historical periodization in the PeriodO period gazetteer, which we describe below.

## THE PERIODO PERIOD GAZETTEER

In their work, archaeologists and historians frequently refer to time periods, such as the "Classical Iberian Period" or the "Progressive Era." These time periods are shorthand representations of commonly referenced segments of time and space. While time periods might have commonly understood definitions, they are typically scattered throughout myriad publications and are treated as shared, assumed knowledge. This leads to difficulty and repeated effort when scholars want to visualize their data in space and over time, which requires mapping these discursive period labels to discrete spatiotemporal ranges (*Rabinowitz, 2014*).

To build the PeriodO gazetteer, we compiled thousands of definitions of time periods from published sources within the fields of archaeology, history, and art history. We mapped these time periods to a consistent data model and published them as linked open data (*Heath & Bizer, 2011*) so that future scholars would be able to link their uses of period terms to information about the provenance of those terms. A web-based faceted browsing interface allows scholars to find and compare period definitions (see Fig. 3), or software developers can use the PeriodO data directly in their own systems. The gazetteer is editable via HTTP; contributors can submit proposed changes in the form of patches, and the PeriodO editors can accept or reject them. All proposed and accepted changes are stored, and each period definition has a history of changes in the form of patch submissions and approvals (*Shaw et al., 2015*). To ease the process of creating patches that conform to the PeriodO data model, we developed an editing interface that runs in a standard web browser (see Fig. 4).

### Data model

PeriodO defines a "period definition" as a scholarly assertion about the name and spatiotemporal extent of a period. The core of a period definition consists of text quoted from the original source indicating the name of the period, its temporal range, and the geographic region to which it applies. Multiple period definitions from the same source are grouped into a period collection. For example, the article "Domestic Architecture and Social Differences in North-Eastern Iberia during the Iron Age (c.525–200 BC)" includes the following sentence:

> For the Catalan area, the complete system with the four above-mentioned categories is not as clearly documented before the fourth century as it is during the Classical Iberian Period (400–200 BC), although differences in the size of the sites, as well as the specialization of the functions of some settlements, can be already detected during the Early Iberian Period (525–400 BC) (*Belarte, 2008*).

This sentence contains two assertions defining period extents, so it is modeled in PeriodO as two period definitions. The first definition has the label "Classical Iberian Period" and its start and end points are labeled as "400 BC" and "200 BC" respectively. The second definition has the label "Early Iberian Period" and its start and end points are labeled as "525 BC" and "400 BC" respectively. The spatial extent of both definitions is labeled as "Catalan area". All of these labels are taken verbatim from the source text and should never change.

Because they come from the same source, these two period definitions are grouped into a period collection. The bibliographic metadata for the source article is associated with this period collection. (In the event that a source defines only a single period, then the period collection will be a singleton.) Belonging to the same period collection does not imply that period definitions compose a periodization. A periodization is a single coherent, continuous division of historical time, each part of which is labeled with a period term. A period collection, on the other hand, is simply a set of period definitions that share the same source. When the period definitions in a period collection *do* compose a periodization, this can be indicated through the addition of statements relating the period definitions to one another, e.g., as belonging to the same periodization and having a specific ordering.

Because source languages, dating systems, and naming of geographical regions can vary widely, labels taken verbatim from source documents are insufficient for indexing and visualizing period definitions in a uniform way. Thus the rest of the PeriodO data model consists of properties added by PeriodO curators to normalize the semantic content of these textual labels. First, all periods originally defined in a language other than English are given an alternate English-language label. When a period definition was originally defined in English, the alternate label may make minor changes for consistency. For example, Belarte's definition of the "Classical Iberian Period" period was given an alternate label of "Classical Iberian", removing the word "Period" for brevity and consistency with other definitions. Next, the specification of temporal start and end points is standardized by adding ISO 8601 lexical representations of proleptic Gregorian calendar years[1]: −0399 for "400 BC" and −0199 for "200 BC". Finally, descriptions of spatial extent are normalized by adding references to "spatial things", typically modern nation-states. In this case both definitions are linked to the spatial thing identified by http://dbpedia.org/resource/Spain. The complete PeriodO representation in Turtle of Belarte's collection of period definitions is given in Fig. 1.[2]

## PERIODO AS LINKED DATA

We have taken pains to make it easy to work with the PeriodO dataset, particularly keeping in mind developers who do not use an RDF-based tool stack. The dataset is published as JSON, which is easily parsed using standard libraries in most programming environments including, of course, web browsers. But while JSON provides an easy and convenient way to work with the PeriodO dataset by itself, we knew that many users would want to combine it with the growing body of scholarly Linked Data being published on

[1] Proleptic refers to dates represented in some calendar system that refer to a time prior to that calendar's creation. The Gregorian calendar was adopted in 1582, but most of our dates fall in years prior to that one.

[2] Turtle is a human-readable syntax for serializing RDF graphs (*Carothers & Prud'hommeaux, 2014*).

```
@prefix bibo: <http://purl.org/ontology/bibo/> .
@prefix dcterms: <http://purl.org/dc/terms/> .
@prefix foaf: <http://xmlns.com/foaf/0.1/> .
@prefix periodo: <http://n2t.net/ark:/99152/p0v#> .
@prefix skos: <http://www.w3.org/2004/02/skos/core#> .
@prefix time: <http://www.w3.org/2006/time#> .
@prefix xsd: <http://www.w3.org/2001/XMLSchema#> .

# Belarte's definition of the Early Iberian Period.
<http://n2t.net/ark:/99152/p06xc6mq829>
    a skos:Concept ;
    skos:prefLabel "Early Iberian Period" ;
    skos:altLabel "Early Iberian Period"@eng-latn, "Early Iberian"@eng-latn ;
    skos:inScheme <http://n2t.net/ark:/99152/p06xc6m> ;
    dcterms:language "eng-latn" ;
    dcterms:spatial <http://dbpedia.org/resource/Spain> ;
    periodo:spatialCoverageDescription "Catalan area" ;
    time:intervalFinishedBy [
        skos:prefLabel "400 BC" ;
        time:hasDateTimeDescription [
            time:year "-0399"^^xsd:gYear
        ]
    ] ;
    time:intervalStartedBy [
        skos:prefLabel "525 BC" ;
        time:hasDateTimeDescription [
            time:year "-0524"^^xsd:gYear
        ]
    ] .

# Belarte's definition of the Classical Iberian Period.
<http://n2t.net/ark:/99152/p06xc6mvjx2>
    a skos:Concept ;
    skos:prefLabel "Classical Iberian Period" ;
    skos:altLabel "Classical Iberian Period"@eng-latn, "Classical Iberian"@eng-latn ;
    skos:inScheme <http://n2t.net/ark:/99152/p06xc6m> ;
    skos:note "Equivalent to Iberian III (450-350 B.C.) and IV (350-200 B.C.) - cf. M. Diaz-Andreu
        & S. Keay, 1997. The Archaeology of Iberia; Dominguez in C. Sanchez & G.R. Tsetskhladze,
        2001. Greek Pottery from the Iberian Peninsula." ;
    dcterms:language "eng-latn" ;
    dcterms:spatial <http://dbpedia.org/resource/Spain> ;
    periodo:spatialCoverageDescription "Catalan area" ;
    time:intervalFinishedBy [
        skos:prefLabel "200 BC" ;
        time:hasDateTimeDescription [
            time:year "-0199"^^xsd:gYear
        ]
    ] ;
    time:intervalStartedBy [
        skos:prefLabel "400 BC" ;
        time:hasDateTimeDescription [
            time:year "-0399"^^xsd:gYear
        ]
    ] .

# The collection of period definitions for linking them to their common source.
<http://n2t.net/ark:/99152/p06xc6m>
    a skos:ConceptScheme ;
    dcterms:source [
        dcterms:isPartOf <http://dx.doi.org/10.1111/j.1468-0092.2008.00303.x> ;
        bibo:locator "page 177"
    ] .

# The journal article from which the period definitions were taken.
<http://dx.doi.org/10.1111/j.1468-0092.2008.00303.x>
    dcterms:title "DOMESTIC ARCHITECTURE AND SOCIAL DIFFERENCES IN NORTH-EASTERN IBERIA DURING THE
        IRON AGE (c.525-200 BC)" .
    dcterms:creator <http://id.crossref.org/contributor/maria-carme-belarte-2mkpvn5eyc7oh> ;
    dcterms:issued "2008"^^xsd:gYear ;

# The author of the journal article.
<http://id.crossref.org/contributor/maria-carme-belarte-2mkpvn5eyc7oh>
    foaf:name "MARIA CARME BELARTE" .
```

**Figure 1** **Turtle representation of a PeriodO period collection containing two period definitions originally published by *Belarte (2008)*.**

the Web. Most of our initial contributors of period definitions work in archaeology, a discipline that has several large, well-curated, interlinked, widely used and well-maintained Linked Data datasets (*Isaksen et al., 2014*). Thus, we take advantage of the recent W3C Recommendation of JSON-LD (*Sporny, Kellogg & Lanthaler, 2014*) to make the PeriodO dataset available as Linked Data. By providing a JSON-LD context for the PeriodO dataset, we have made it usable within an RDF-based stack.

## RDF vocabularies

The JSON-LD context maps relationships between PeriodO entities to terms from RDF vocabularies. Of these, the most important is SKOS (*Hobbs & Pan, 2006*). The human-readable labels for a PeriodO definition are mapped to the SKOS `prefLabel` and `altLabel` properties, implying that a PeriodO period definition can be interpreted as a SKOS Concept. The relationship between a period definition and the period collection to which it belongs is mapped to the SKOS `inScheme` property, implying that a period collection is a SKOS `ConceptScheme`. The relationship between a period collection and its source is mapped to the DCMI `source` term, and the various properties in the bibliographic description of the source are mapped to their own appropriate DCMI terms. Finally, the relation between a period definition and its geographical extent is mapped to the DCMI `spatial` term.

The relationships between a period definition and the start and end of its temporal extent are respectively mapped to the OWL-Time `intervalStartedBy` and `intervalFinishedBy` properties. This implies that a period definition, in addition to being a SKOS Concept, is an OWL-Time `ProperInterval` (an interval of time having non-zero duration). Importantly, it also implies that the start and end of a period definition's temporal extent are themselves `ProperIntervals`, not points or instants. This is important because the beginnings and endings of historical periods can never be precisely determined. In the example of the Classical Iberian Period given above, both the beginning and the end of the period are interpreted as intervals with a duration of one year. Interpreting period starts and ends as `ProperIntervals` allows us to make a distinction between the intervals themselves and their descriptions: though the intervals themselves are not precisely specifiable, we can create pragmatic OWL-Time `DateTimeDescriptions` of them for the purposes of comparison and visualization.

The start and end of a period definition's temporal extent are themselves intervals with their own starts and ends, so temporal extent can be associated with a maximum of four values. This is interoperable with other proposed representations of fuzzy, imprecise, or uncertain temporal extents, such as the four `start`, `stop`, `earliest`, `latest` keys proposed for GeoJSON-LD (*Gillies, 2015*). In the current PeriodO data set these four properties only have (ISO 8601) year values, because none of our sources specified endpoints at a more granular level than year. However, we expect to have finer-grained values as we add periodizations of more recent history. At that point we will need to decide upon a unit of representation that makes it simple to compare intervals defined at different levels of granularity. Adding complexity to time interval expressions will be

possible without changing our underlying data model because of the flexibility of our current approach.

The *start*, *latest start*, *earliest end*, *end* approach enables us to represent the most common patterns for defining periods found in our sources. For example a period defined as starting "3000 B.C. (±150 years)" and ending "about 2330 B.C." can be represented with three values: −3149, −2849, and −2329. *Kauppinen et al. (2010)* propose defining curves over intervals to represent fuzziness, imprecision, or uncertainty in order to maximize precision and recall with respect to temporal relevance judgments made by experts. We have chosen not to support such more complex representations at this time because we are focused primarily on representing periods as defined in textual sources. Natural language is already a compact and easily indexable way to represent imprecision or uncertainty. Rather than imposing an arbitrary mapping from natural language to parameterized curves, we prefer to maintain the original natural language terms used. However if scholars begin defining periods with parameterized curves (which is certainly possible) then we will revisit this decision.

### Modeling provenance

To model the provenance of period assertions, we used the Provenance Ontology (*McGuinness, Lebo & Sahoo, 2013*). We record each change to the dataset (a patch) as a `prov:Activity`. This Activity has `prov:startedAtTime` and `prov:endedAtTime` values representing timestamps when the patch was sent and accepted, respectively. The Activity additionally has two `prov:used` statements: one which refers to the specific version of the entire dataset to which the patch was applied (for example, http://n2t.net/ark:/99152/p0d?version=1), and one referring to the patch itself as a `prov:Entity`. The patch Entity contains a URL to the JSON-Patch file which resulted in the change Activity (*Nottingham & Bryan, 2013*). Finally, the Activity has `prov:generated` statements for each of the period collections and period assertions (implied to be of the type `prov:Entity`) that were affected by the given patch. Each of these affected entities has a `prov:specializationOf` statement that refers to the permanent identifier for the period assertion or collection (with no particular version specified). If the affected entities are revisions of an existing entity, they have `prov:wasRevisionOf` statements that refer to the version that they were descended from.

We publish a changelog at http://n2t.net/ark:/99152/p0h#changelog that represents the sequential list of `prov:Activity` entities that created the current version of the dataset as an ordered RDF list. In this way, one can reconstruct the origin of each change to the dataset as a whole, or to individual period assertions.

### Minting long-term URLs

In addition to mapping relationships to well-known vocabularies, interpreting PeriodO as Linked Data requires a way to assign URLs to period collections and definitions. As shown in Fig. 1, period definitions and period collections in the dataset are given short identifiers: `p06xc6mvjx2` identifies the definition of the Classical Iberian Period, and `p06xc6m` identifies the collection to which it belongs. But these identifiers are only useful

within the context of the PeriodO dataset; they are not guaranteed to be unique in a global context and, unless one already has the PeriodO data, one cannot resolve them to obtain representations of the entities they identify. URLs, on the other hand, are globally unique and can be resolved using HTTP to obtain representations; this is the core concept behind Linked Data. So, we need a way to turn the short PeriodO identifiers into URLs.

To turn PeriodO identifiers into URLs we rely on the ARK identifier scheme (*Starr et al., 2012*) provided by the California Digital Library (CDL). First, we include in the JSON-LD context a `@base` value specifying the base URI (http://n2t.net/ark:/99152/p0) to use when interpreting the PeriodO dataset as Linked Data. This allows the short PeriodO identifiers to be interpreted as URLs; for example `p06xc6mvjx2` is interpreted as a relative reference to the URL http://n2t.net/ark:/99152/p06xc6mvjx2. The hostname of this URL (`n2t.net`) is the registered name of the CDL's Name-to-Thing resolver, which is similar to other name resolution services for persistent URLs such as PURL. We have registered with the EZID service a single ARK identifier (`ark:/99152/p0`), providing them with the URL of the HTTP server currently hosting the canonical PeriodO dataset. Thus any request to a URL starting with http://n2t.net/ark:/99152/p0 will be redirected to that server. An HTTP GET to http://n2t.net/ark:/99152/p0d.jsonld will return the entire dataset, while GETting (for example) http://n2t.net/ark:/99152/p06xc6mvjx2.jsonld will return a JSON-LD representation of Belarte's definition of the "Classical Iberian Period."

## PERIOD ASSERTIONS AS NANOPUBLICATIONS

We created the PeriodO dataset based on the same core concerns of nanopublication authors: to extract, curate, and publish small, computable concepts from their broader sources while still preserving their provenance. A nanopublication is made up of an assertion, the provenance of that assertion, and the provenance of the nanopublication itself. In PeriodO, these are:

- **Assertion**: the definition of a period.
- **Provenance**: the source this period was derived from. This may be a citation of a printed work or a URL for a resource hosted on the web.
- **Provenance of nanopublication**: the history of the period definition within the PeriodO system, including the date it was added or changed, the identity of the person who submitted or changed it, and the identity of the person who approved additions or changes.

Figure 1 shows two period definitions with the same provenance. Each of these definitions is represented by an individual nanopublication. The nanopublication for the "Early Iberian Period" is shown in Fig. 2. While PeriodO period definitions readily map to the nanopublication scheme, we faced several challenges during our creation of the dataset due to its interpretive nature.

### The unfalsifiable nature of time period definitions

The current version of the Nanopublication Guidelines includes a note suggesting that the guidelines be amended to state that an assertion published as a nanopublication should be

```
@prefix : <http://n2t.net/ark:/99152/p06xc6mq829/nanopub1#> .
@prefix bibo: <http://purl.org/ontology/bibo/> .
@prefix dcterms: <http://purl.org/dc/terms/> .
@prefix foaf: <http://xmlns.com/foaf/0.1/> .
@prefix np: <http://www.nanopub.org/nschema#> .
@prefix periodo: <http://n2t.net/ark:/99152/p0v#> .
@prefix prov: <http://www.w3.org/ns/prov#> .
@prefix skos: <http://www.w3.org/2004/02/skos/core#> .
@prefix time: <http://www.w3.org/2006/time#> .
@prefix xsd: <http://www.w3.org/2001/XMLSchema#> .

# A graph of statements identifying the nanopublication and its parts.
:head {
    <http://n2t.net/ark:/99152/p06xc6mq829/nanopub1>
        a np:Nanopublication ;
        np:hasAssertion :assertion ;
        np:hasProvenance :provenance ;
        np:hasPublicationInfo :pubinfo .
}

# A graph with the assertions being made (i.e. the period being defined).
# This is identical to the representation of the first period definition in Figure 1.
:assertion {
    <http://n2t.net/ark:/99152/p06xc6mq829>
        a skos:Concept ;
        skos:prefLabel "Early Iberian Period" ;
        skos:altLabel "Early Iberian Period"@eng-latn, "Early Iberian"@eng-latn ;
        skos:inScheme <http://n2t.net/ark:/99152/p06xc6m> ;
        dcterms:language "eng-latn" ;
        dcterms:spatial <http://dbpedia.org/resource/Spain> ;
        periodo:spatialCoverageDescription "Catalan area" ;
        time:intervalFinishedBy [
            skos:prefLabel "400 BC" ;
            time:hasDateTimeDescription [
                time:year "-0399"^^xsd:gYear
            ]
        ] ;
        time:intervalStartedBy [
            skos:prefLabel "525 BC" ;
            time:hasDateTimeDescription [
                time:year "-0524"^^xsd:gYear
            ]
        ] .
}

# A graph of statements about the provenance of the assertions.
:provenance {
    :assertion dcterms:source [
        dcterms:isPartOf <http://dx.doi.org/10.1111/j.1468-0092.2008.00303.x> ;
        bibo:locator "page 177"
    ].

    <http://dx.doi.org/10.1111/j.1468-0092.2008.00303.x>
        dcterms:creator <http://id.crossref.org/contributor/maria-carme-belarte-2mkpvn5eyc7oh> ;
        dcterms:issued "2008"^^xsd:gYear ;
        dcterms:title "DOMESTIC ARCHITECTURE AND SOCIAL DIFFERENCES IN NORTH-EASTERN IBERIA DURING
            THE IRON AGE (c.525-200 BC)" .

    <http://id.crossref.org/contributor/maria-carme-belarte-2mkpvn5eyc7oh>
        foaf:name "MARIA CARME BELARTE" .
}

# A graph of statements about the provenance of the nanopublication itself.
:pubinfo {
    <http://n2t.net/ark:/99152/p06xc6mq829/nanopub1> prov:wasGeneratedBy <p0h#change-1> ;
        prov:generatedAtTime "2015-07-29T21:49:31"^^xsd:dateTime ;
        prov:wasAttributedTo <http://orcid.org/0000-0002-3617-9378> .
}
```

**Figure 2** Nanopublication of *Belarte (2008)*'s definition of the "Early Iberian Period".

"a proposition that is falsifiable, that is to say we can test whether the proposition is true or false" (*Groth et al., 2013*). Were this amendment to be made, PeriodO nanopublications would be in violation of the guidelines, as period definitions in PeriodO, like most of the information produced in the humanities, are neither testable nor falsifiable. Consider the

assertion "there is a period called the Late Bronze Age in Northern Europe, and it lasted from about 1100 B.C. to 500 B.C." The "Late Bronze Age" is a purely discursive construct. There was no discrete entity called the "Late Bronze Age" before it was named by those studying that time and place. Consequently, one cannot disprove the idea that there was a time period called the "Late Bronze Age" from around 1100 B.C. to 500 B.C.; one can only argue that another definition has more credence based on non-experimental, discursive arguments.

The proposed falsifiability requirement makes sense in certain contexts. Computational biologists, for example, wish to connect, consolidate, and assess trillions of measurements scattered throughout a rapidly growing body of research findings. Their goal is to create a global, connected knowledge graph that can be used as a tool for scientists to guide new discoveries and verify experimental results. In the PeriodO context, however, we are not concerned with making an exhaustive taxonomy of "correct" periods or facilitating the "discovery" of new periods (a non sequitur—there are no periods that exist in the world that are awaiting discovery by some inquiring historian or archaeologist). Instead we are interested in enabling the study and citation of how and by whom time has been segmented into different periods. It is not necessary that these segmentations be falsifiable to achieve this goal; they only need to be comparable.

*Kuhn et al. (2013)* expressed concern that requiring formal representation for all scientific data published as nanopublications "seems to be unrealistic in many cases and might restrict the range of practical application considerably." Similarly, requiring assertions to be unambiguous and falsifiable would unnecessarily restrict the practical application of nanopublication. The nature of nanopublication assertions should ultimately be determined by the practical needs of the researchers who use them. What is important about nanopublications is not the nature of the assertions, but the expression of provenance. Provenance is particularly important for non-scientific datasets, since the assertions made are so dependent on their wider discursive context. When assertions cannot be tested experimentally, understanding context is critical for judging quality, trustworthiness, and usefulness.

## The critical role of curation

Another difference between the PeriodO dataset and traditional nanopublications is the unavoidable curatorial work necessary to extract practically useful assertions from textual period definitions. In all of the applications of nanopublications we found, the published assertions typically appeared in the form of measurements or well-defined relationships between discrete entities. These are types of data which humans or computers can easily and reliably extract from research findings. Our dataset, in contrast, required explicit curatorial decisions: a time period exists within a certain spatiotemporal context, and there is no sure way to discretely, accurately, and unambiguously model such boundaries. While a human might be able to have a nuanced understanding of temporary and ever-shifting political boundaries or the uncertain and partially arbitrary precision suggested by "around the beginning of the 12th century BC," we cannot assume the same of computers.

Therefore, in order for our dataset to be readily algorithmically comparable, we had to map discursive concepts to discrete values. Our curatorial decisions in this regard reflect a compromise between uniformity, potential semantic expressiveness, and practical usefulness.

As humanities scholars publish their own nanopublications (or linked data in general), they will go through similar curatorial processes due to the interpretive, unstandardized nature of humanities datasets discussed above. There is a temptation in this process to imagine perfect structured descriptions that could express all possible nuances of all possible assertions. However, chasing that goal can lead to overcomplexity and, in the end, be practically useless. In describing period assertions as linked data, we adopted a schema that was only as semantically complicated as was (a) expressed in our collected data and (b) necessitated by the practical needs of our intended users. As we started to collect data, we considered the basic characteristics of a dataset that would be necessary to accomplish the retrieval and comparison tasks that our intended users told us were most important. These tasks included:

- Finding how the definition of periods have differed across time/authors, or finding contested period definitions. ("How have different authors defined the Early Bronze Age?")
- Finding all periods within a certain span of time. ("What time periods have been used to describe years between 100 AD to 500 AD?").
- Finding all periods within a certain geographic area. ("What time periods have scholars used in Northern Europe?")
- Finding periods defined for different languages. ("What time periods have been defined in Ukranian?")

Figure 3 shows how these various tasks can be completed using the faceted browsing interface to the PeriodO dataset. Implementing this interface required imposing consistency upon how we represented the temporal and spatial coverage of period definitions, even though this consistency does not exist in the original sources.

Our initial approach to imposing consistency on temporal extents was to express the termini of periods as Julian Days represented in scientific notation. Julian Days are a standard form of time measurement commonly used by astronomers to represent dates in the far historical past. Julian Days work by counting the number of continuous days that have passed since January 1, 4713 BC in the Proleptic Julian calendar. Conceptually, this is a similar measurement to the common Unix time standard, which counts the number of milliseconds that have passed since midnight GMT on January 1, 1970. The idea is that by counting forward using well-defined units since an accepted epoch, one can escape the inconsistencies and periodic lapses that characterize different calendrical systems. Representing Julian Days using scientific notation allows one to express variable levels of uncertainty. See examples of this notation system in Table 1.

However, in practice, we found this scheme to be overly complex. The imposition of a level of uncertainty, while theoretically useful in certain cases, was often not appropriate.

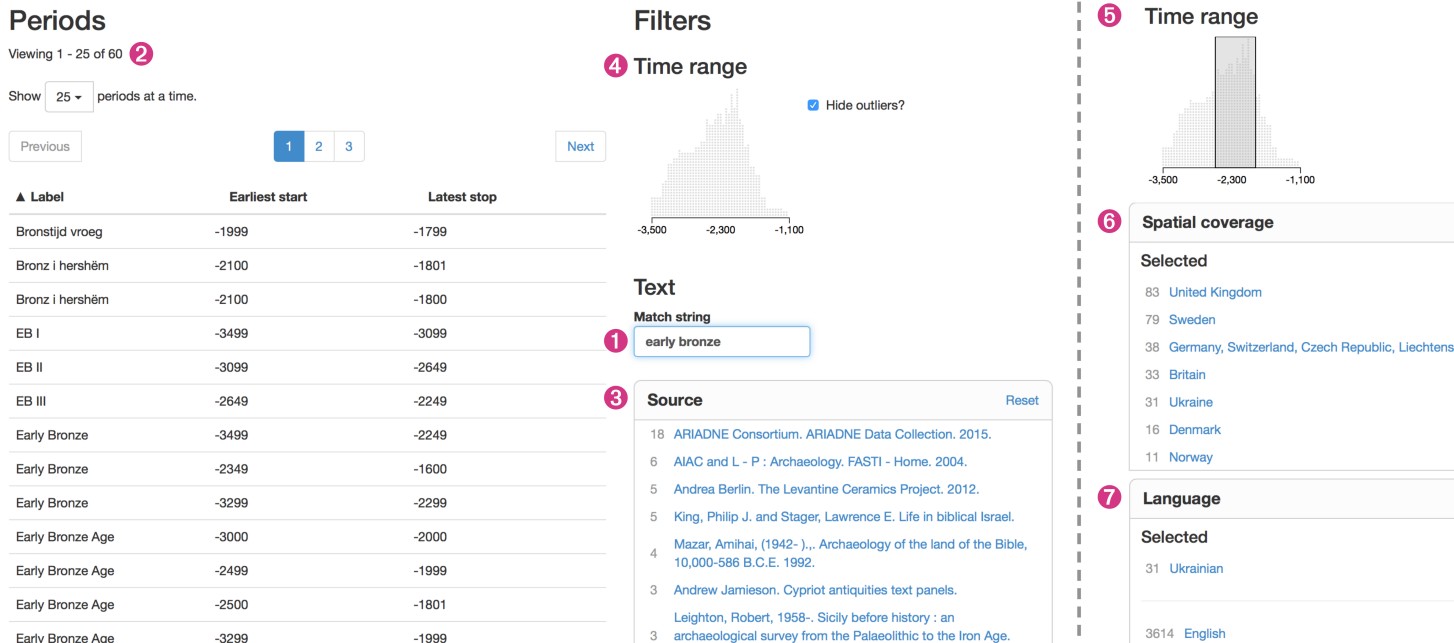

**Figure 3** **Finding and comparing period definitions in PeriodO.** Searching for "early bronze" (1) results in sixty period definitions with matching labels (2), from a variety of sources (3). The time range facet (4) updates to show the distribution of temporal extents defined by these various sources. Users can query for period definitions with temporal extents within a specific range of years using the time range facet (5), period definitions with spatial extents within a named geographic area using the spatial coverage facet (6), or period definitions in specific languages using the language facet (7). Queries may combine values from any of these facets.

**Table 1** **Example scientific notation of Julian days.**

| Scientific notation | Julian day (JDN) | Proleptic gregorian |
|---|---|---|
| 1.3E6 | Between JDN 1,250,000 and JDN 1,350,000 | 1150 BC $\pm$ 150 years |
| 1.30E6 | Between JDN 1,295,000 and JDN 1,305,000 | 1150 BC $\pm$ 15 years |
| 1.300E6 | Between JDN 1,299,500 and JDN 1,300,500 | 1150 BC $\pm$ 1.5 years |

In almost every single case that we observed, authors did not explicitly state a precise level of uncertainty for their temporal expressions. By adding precise uncertainty ourselves, we would, in effect, have been putting words in authors' mouths. Further, Julian Days are not widely used outside of very specific disciplines, meaning that consumers of our data would have to convert to a more familiar time system before being able to understand or use our data. Instead of the Julian Day model, we settled on the four-part ISO date schema, described above. This model is less expressive for complicated forms of uncertainty, but it is less complex and more easily understood by both our target audience and typical software programs. ISO dates were simple to convert to, since nearly all of the period assertions we observed were drawn from sources based on Western calendars. If and when we encounter period definitions that require more complex time expressions or are based on varying calendrical systems, we will revisit the question of whether the four-part scheme is sufficient.

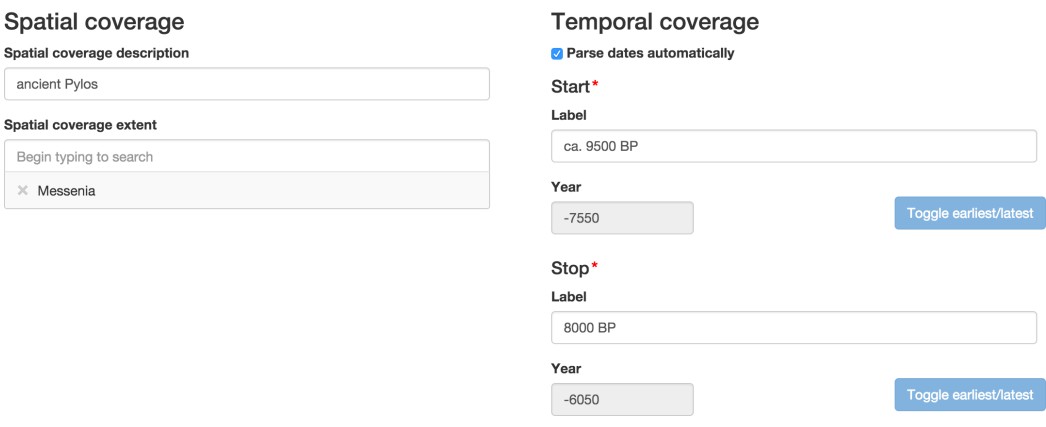

**Figure 4** **Part of the interface for editing period definitions.** Labels for temporal extent boundaries are taken verbatim from the source, entered as free text, and automatically parsed into ISO 8601 year representations. Labels for spatial coverage are entered as free text, and using an autocompletion interface the user can specify the modern-day administrative units (e.g., nation-states) that approximate this spatial coverage.

To encourage a consistent representation of temporal extent for all period definitions, we built a simple grammar and parser for date expressions that covered the vast majority of our sample data. The parser takes in a string like "c. mid-12th century" and outputs a JSON string consistent with our data model. It can also produce naïve interpretations of descriptions like "mid-fifth century," assigning them to the third of the epoch described according to the conventional segmentation of "early," "mid," and "late," "Mid-fifth century" would, then, be parsed as the range of years 401–434. The parser is intended to be used interactively, as a generator of suggestions for standard ways to represent certain forms of time description. To keep the quality of the gazetteer high, we do not intend for the parser to be used to fully automatically "extract" period definitions from texts. Similarly, we created an autocomplete interface to modern political entities to allow users to enter spatial coverage. These interface components help curators produce a practical approximation of spatiotemporal coverage rather than a complete, unambiguous representation. The interface we created to allow users to add and edit period definitions is shown in Fig. 4.

## PROJECT STATUS AND FUTURE WORK

As of late 2015, we have gathered just over 3,500 period definitions from 78 sources, including monographs, journal articles, and online databases. Each period has been assigned a permanent URL, which can be resolved to view its definition and provenance as HTML, JSON-LD, or Turtle. Several projects have begun to use our gazetteer to add spatiotemporal information to their work, including the Open Context research data repository (http://opencontext.org), the ARIADNE archaeological research data infrastructure project (http://ariadne-infrastructure.eu), and the Portable Antiquities Scheme database of archaeological finds in the UK (https://finds.org.uk).

As more projects begin to integrate PeriodO identifiers for time periods, we hope to gather information on their citation and use. This would include both studying the historical use of attributed period definitions as well as tracking the citation of PeriodO period identifiers going forward. Such a study would allow us to observe how periods come into circulation and fall out of favor. Tracing the connections fostered by use of our gazetteer would demonstrate the potential benefits of a linked data approach in the humanities.

We are also in the process of reaching out to period-defining communities beyond classical archaeology and ancient history. We expect that this will require some extensions of and revisions to the current PeriodO data model. First, as we begin to collect definitions of periods closer to the present, we expect to extend our model of temporal extent to allow for more fine-grained interval boundaries than years. This will require a unit of representation that allows comparisons between intervals defined at different levels of granularity. (The approach based on Julian Days, described in Table 1, may be useful for this.) Second, as we begin to include more non-Western period definitions, we will need to ensure that we can still map years to ISO 8601 representations. At the very least, this will require extending the temporal expression parser, and it may require changes to the data model as well, for example to state explicitly the calendar system used by the original authors. Finally, as more historians begin publishing their work as datasets or software, we may begin to encounter periods defined not in natural language but using some formalism, such as the curves proposed by *Kauppinen et al. (2010)*. These will require us to find a way of including these formalisms directly in our definitions.

## CONCLUSION

As scholars of all disciplines continue to integrate computational methods into their work, the need to preserve provenance will only become more important. This is as true in the humanities and social sciences as it is in the natural sciences. Nanopublication is an useful way to locate the production of "data" within a wider scholarly context. In this way, it echoes old ideas about hypertext which were concerned with relations of provenance, authorship, and attribution (*Nelson, 1999*). The PeriodO period gazetteer shows that this approach is relevant and feasible even to fields outside of the experimental, observable sciences.

### Funding

This work was generously funded by a Digital Humanities Start-Up Grant from the National Endowment for the Humanities (grant number HD-51864-14). The funders had no role in study design, data collection and analysis, decision to publish, or preparation of the manuscript.

### Grant Disclosures

The following grant information was disclosed by the authors:
National Endowment for the Humanities: HD-51864-14.

## Competing Interests

The authors declare there are no competing interests.

## Author Contributions

- Patrick Golden and Ryan Shaw conceived and designed the experiments, performed the experiments, analyzed the data, contributed reagents/materials/analysis tools, wrote the paper, prepared figures and/or tables, performed the computation work, reviewed drafts of the paper.

## Data Availability

PeriodO dataset: http://n2t.net/ark:/99152/p0.

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
