# Peer review of "Nanopublication beyond the sciences: the PeriodO period gazetteer"

_PeerJ Computer Science, doi:10.7717/peerj-cs.44_

## Round 0.1 · original submission · Major Revisions

Please see the comments of reviewers.

Please attend to those, including the suggestion for title change.

Generally the writing is good, however, there is some repetition that should be eliminated. This is especially annoying when repeated comments just assert the importance of nanopublication; these sound rather polemical.

One big question, requiring explanation, is about the formalism.

Having long figures without description seems just to consume space. What would a scholar learn from these?
If they are included, there should be some justification and goal achieved. Further, there is the assumption that RDF is a good idea, with no discussion, and no evidence given of the pros and cons, or of the limitations or benefits.

In general this is rather short and lacking in detail. Some sections are very short, and should be expanded:
* Uses starting on p. 2, line 66
* Future Work starting on p. 11, line 533

Some explanation is needed for the illustrations.
* Figs. 1, 2 could have the various parts explained.
* Fig. 3 should be explained in detail

Additional examples would be particularly helpful

It also is unclear if historical documents can be automatically analyzed through information extraction methods, or if the only way to submit data is using the editor for each entry, as in Fig. 3.

Following are more specific comments:

The status of PeriodO is not clear but should be.
How big is it?
What data is included?
Who uses it?
What are the uses so far?
How does one use it if interested?
What is the software architecture, ways to access it?
There is mention on p. 10, lines 494-499, of hypothetical queries, but it is not clear if these are real examples, nor how the queries would be entered, nor what results would come from them.

Page 7, line 309: fill in the [cite]
line 319: unclear re “at no particular version”
line 322: he -> the

References
* Buckland has 0(0)
* Hobbs says “report” but gives no report number nor URL
* Meeks gives no publisher nor URL
* Sporny says “report” but gives no report number nor URL

Reviewer 1 ·

Basic reporting

The PeerJ Computer Science policies list is quite lengthy. I do not feel that I can determine whether the submission is in compliance or meets all of the requirements. I will proceed with the review as if the submission meets these.

Experimental design

The manuscript describes application of a rigorous methodology to structuring and organizing data meant for humanistic study. As such, it does not mimic or pretend to conform to scientific methods used in quantitative or qualitative research. However the data model described in the manuscript can be valuably applied to collections scientific data. The attributes of nanopublication have great value over traditional publications in journals - either those reporting on scientific work or humanities research. Gregory Crane has noted in numerous white papers and presentations the advantages of "recombinant" documents. Carole Goble has spoken frequently about the need for "staged-release" of findings of scholarly workflows. Nanopublication as used here gives a useful formalization of these traits.

Validity of the findings

The authors address, correctly in my view, some of the inherent difficulties of working with humanities data. Spatial and temporal information associated with a given place at a given time can often not be established with accuracy. Geographical locations of places vary - different texts and maps move them about. The names of the places change as well. This is sometimes a result of the ambiguity of natural language and translation possibilities. Temporal data is subject to the same uncertainties. The work described in this manuscript is very important in that regard and the authors convincingly make a case that humanistic

Linked data and semantic web technologies show their promise and power in humanities research. The authors recognize this and make a strong case for a liberal interpretation of data as used by humanists.

Additional comments

The paper elevates the discussion of linked data and humanities research by providing a clear description of the nature of humanities data and reasonabe expectations in its use. Provenance will always be a problem. The exact origin of assertions and factual data may be beyond one's reach or simply unknowable. The tasks of data curation and stewardship have not been yet identified clearly or who will be responsible for them. It may be too much to ask of the scholar/researcher for certain types of studies. This topic deserves a separate study. I did not see a need to include all the detail in the that section of the paper.

·

Basic reporting

NBS is a challenging article for the uninitiated. As a scholarly communications librarian I am not the typical PeerJ Computer Science reader. However, I really enjoyed the challenge after finding nanopub.org, perio.do, and reading Mons et al. (2011) “The value of data.” My sticky-note comments target some of the places in the article where examples would have been helpful.

Experimental design

The first paragraph of the conclusion would have provided me with a better introduction to the article.

Unfalsifiable definitions and the role of curation are hugely important issues to the humanities and social sciences. I hope they will have the opportunity to read this article.

Validity of the findings

The authors appropriately explained the benefits of nanopublications to the humanities and social sciences while not denying the areas where PeriodO doesn't exactly fit the model.

This article deserves to join the scientific literature.

Additional comments

I would have gotten more out of the first reading of this article if the explanatory information had preceded the technical details. I would entitle this article “Nanopublication Beyond the Sciences and the PeriodO Case Study” since the authors did not offer any other examples of nanopubs in the humanities.

---

## Round 0.2 · Minor Revisions

Thanks for carefully attending to comments.
Below are some copy editing type changes that should be easy to make, and then this should be all set for publication.
Nice work.
- - - specifics follow- - -
Page 4, line 180: make make -> make

Page 10, Figure 3:
* It appears things are cut off on the left.
* Top left has “ime.” Thus something preceding it was cut off.
* There is no red circle with “2” but the caption refers to (2).

Page 12, Figure 4: Why not change, for consistency:
Spatial Coverage Extent -> Spatial coverage extent
Also, the line below
x Messenia
seems to be missing something to label it?

Please see if you can reduce the number of occurrences of “also”.
In particular, try to avoid such use in consecutive sentences.

References:
line 439 Carothers et al.: turtle -> Turtle
line 484 Nottingham and Bryan: Ietf -> IETF
line 491 Sporny et al. ld -> LD

·

Basic reporting

The authors took the reviewers comments seriously and have greatly improved the readability of this article.

Experimental design

No comments.

Validity of the findings

No comments.

---

## Round 0.3 · accepted · Accept

The requested issues were addressed, so this article is now Accepted. Congratulations.